# Hybrid Bipedal Locomotion Based on Reinforcement Learning and Heuristics

**DOI:** 10.3390/mi13101688

**Published:** 2022-10-07

**Authors:** Zhicheng Wang, Wandi Wei, Anhuan Xie, Yifeng Zhang, Jun Wu, Qiuguo Zhu

**Affiliations:** 1Institute of Cyber-Systems and Control, Zhejiang University, Hangzhou 310027, China; 2Zhejiang Lab, Hangzhou 311121, China; 3Tianjin Jiazi Robot Technology Co., Ltd., Tianjin 300450, China; 4State Key Laboratory of Industrial Control Technology, Zhejiang University, Hangzhou 310027, China

**Keywords:** legged locomotion, reinforcement learning, humanoid, motion planning

## Abstract

Locomotion control has long been vital to legged robots. Agile locomotion can be implemented through either model-based controller or reinforcement learning. It is proven that robust controllers can be obtained through model-based methods and learning-based policies have advantages in generalization. This paper proposed a hybrid framework of locomotion controller that combines deep reinforcement learning and simple heuristic policy and assigns them to different activation phases, which provides guidance for adaptive training without producing conflicts between heuristic knowledge and learned policies. The training in simulation follows a step-by-step stochastic curriculum to guarantee success. Domain randomization during training and assistive extra feedback loops on real robot are also adopted to smooth the transition to the real world. Comparison experiments are carried out on both simulated and real Wukong-IV humanoid robots, and the proposed hybrid approach matches the canonical end-to-end approaches with higher rate of success, faster converging speed, and 60% less tracking error in velocity tracking tasks.

## 1. Introduction

In recent years, various legged robots are developed for traversing through rough and dangerous terrains and working as a replacement for human effort [1]. Specially, dynamic locomotion control, as an essential part of a robust and agile legged robot, has been studied from various aspects.

Model-based methods are first studied and developed [2]. Heuristics methods are widely adopted during early exploration, Raibert proposed hopping controller [2] and extended to bipedal and quadruple robots, and Pratt proposed Virtual Actuator Control [3] and Virtual Model Control (VMC) [4] on bipedal locomotion tasks. These methods adopt reduced modeling and abstract the legged robot into a linear inverted pendulum (LIP) [5] and its derivatives. To further improve the decision, optimization-based methods are introduced to locomotion control [6]. To prevent redundant iterations when searching for the optimal solution, Bledt et al. proposed a regularization-based approach [7]. It uses sub-optimal heuristics to inform the solver through cost function. As a low-cost policy that can approximate optimal solutions with linearization around the working limit cycle, heuristics are still valuable.

Model-based methods have made great advances and are tested to be precise, predictive, and adaptive in various scenes. As they are all based on simplified models that guarantee convergence and simplicity, the accuracy and performance drop when being far away from the working point.

As an emerging and model-free technology, deep reinforcement learning (DRL) has been a popular research field. Previous studies have demonstrated its potential for locomotion tasks. Haarnoja and Tan [8,9] first utilized learning-based gaits on a real robot and verified the feasibility of the end-to-end route. Subsequently, the work in [10,11,12] presents learning separate skills such as trotting and fall recovery using a similar framework. As for bipedal locomotion, a harder task, Hurst [13,14,15] proposed special periodical switching rewards and achieved robust locomotion over planes and stairs. To recombine multiple motor skills into an adaptive multi-modal policy within one end-to-end framework, Yang [16] designed a switching mechanism guided by a phase indicator. Based on that, the multi-expert learning architecture [17] was proposed. It fuses multiple NNs into one according to a gating network’s output to produce adaptive behaviors in response to changing situations. Meanwhile, expanding information sources with perceptual sensing such as heightmap scanning [18] and introducing latent encoders are also means to enhance trained policies. ANYmal robot conquered a series of challenging terrains with privilege learning and latent encoder [19,20]. Kumar et al. proposed Rapid Motor Adaptation framework [21] that enables latent space identification and realized quadrupedal locomotion without predefined references or trajectory generators.

On basis of end-to-end frameworks where the action is completely learned, the sum of model-based control signals and the action of learned policies provide external guidance and avoid pointless blind exploration during training. Low-cost model-based controllers such as central pattern generator [19,22,23], model-based gait library [24,25,26], and heuristic references [27,28] are often adopted in these approaches to achieve agile real-time controlled locomotion. In these frameworks, NN policies learn the residual between optimal decision and reference given by model-based modules. A common issue facing such a summation mechanism is that learned policy might conflict with the other component and cause rigid policy constrained by reference, or the learned policy overwrites the other [22]. Both parts counteract, resulting in a mediocre final performance.

The most similar methodology to ours is linear policy locomotion developed by Krishna [29,30], where the biped’s swing leg is controlled by simple NN policy, and the stance leg is controlled by a model-based controller. However, we have an opposite understanding of the traits of walking phases. In previous studies, we observed that the stance leg performs agile movements through acting ground reaction force (GRF) instead of acting specific joint position trajectory. The swing leg works with a relatively low payload and without restriction from the ground, but the foothold decided by it significantly impacts the subsequent body states. Thus, joint-level accuracy and higher stiffness of swinging control are required. Swing control is a rather deterministic tracking task and can be accomplished by heuristic method and similar control policies with fixed-based manipulators [31]. Meanwhile, the stance leg faces more uncertainty during interaction with terrain and is often controlled by torque controllers [25]. Adopting a learning-based method on stance control can improve robustness compared with model-based approaches. It also narrows down the space for exploration and grants higher efficiency in training.

In pursuit of better performance, separated controllers should be activated during the corresponding phase and adapt to the phase’s attribute, instead of applying a unified controller to fulfill different demands. Inspired by such philosophy, the paper proposes a novel conflict-free hybrid approach to generate adaptive bipedal locomotion with a higher success rate and is easy to transplant to other biped robots with different joint configurations since the policy produces GRF commands. The proposed method is computationally efficient, and also light-weight since there is no iteration-based numerical solver involved in the framework. The main contributions of this paper are as follows:We proposed an efficient hybrid locomotion policy that divides the task into a stance part using model-free learning-based stance control and a swing part using heuristics swing control. The trained policy can perform controllable and regulated gait.To prevent policy divergence caused by fierce randomization at early stages, we proposed a parallel curriculum learning schema with two-stage progressive domain randomization for challenging locomotion tasks.To further reduce the magnitude of sensor randomization which withdraws training speed, we proposed heuristic-based regularization feedback loops on real-world robot systems in addition to a two-branch hybrid controller for resisting simulation model mismatch and assisting stabilization.

The organization of this paper is as follows: Section 2 describes the hardware and software platform and notations. Section 3 introduces the overall structure of our method. Section 4 presents simulation and deployment results and comparisons with an end-to-end benchmark. Finally, Section 5 states the discussion and inspiration for future work.

## 2. Platform and Notations

### 2.1. Robot Model

The robot platform adopted in our work is Wukong-IV adult-size humanoid. Wukong-IV humanoid is designed and built by our research team. It is 1.4 m tall and weighs 45 kg, actuated by electric motor joints. The robot has 6 degrees of freedom (DoF) on each leg and 4 DoFs on each arm. As a bionic humanoid robot, Wukong-IV’s joint configuration and mass distribution bare resemblance to biological humans. A picture of its appearance is shown in Figure 1A, the articulated system model used in simulation training is shown in Figure 1B, and more detailed specifications are listed in Table A1.

The simulation environment is built with RaiSim [32]. Policy neural networks and their training algorithm are utilized with PyTorch [33]. All training is performed on a desktop workstation with a central processor of Intel Xeon Gold 6242R and a graphic processor of NVIDIA Geforce RTX 3090.

### 2.2. Math Notations

In this article, the state of WuKong-IV’s floating base is denoted by $q_{b}$ = {$p_{x}$, $p_{y}$, $p_{z}$, ψ, θ, ϕ}. The first three elements compose base position, also denoted by *p*. The last three are roll, pitch, and yaw angle of base. Robot base quaternion is denoted by ξ. Base linear and angular velocity is *v* and ω. The joint level state is composed of joint angle *q* and joint angle velocity $\dot{q}$. In stance control, the policy gets input $o_{t}$ and outputs action $a_{t}$. The swing foot’s position in the robot frame is denoted by *e*. Joint torques of legs are denoted by $\tau_{stance}$ and $\tau_{swing}$. To represents which foot touches the ground, gait is denoted by *c*.

For all notations, ${\left(\cdot \right)}_{t}$ means the quantity in time step *t*, ${\left(\cdot \right)}^{*}$ means the expected value of the quantity, and δ(·) means compensation to be added to the scripted quantity.

## 3. Materials and Methods

The control frame of hybrid locomotion policy can be described in Figure 2. There are three main branches in the overall structure. The middle branch (blue blocks) shows the learning-based stance control policy (Section 3.1). The network outputs the expected GRF to the stance leg. GRF maps to joint torque via jacobian and then is acted by the robot. The upper branch (orange blocks) shows the heuristic swinging control policy (Section 3.3) which is a position-based method inspired by capture point method. The next foot placement of swing legs is calculated from a clock signal and robot state, then cosine interpolation generates the air position of swing foot, then inverse kinematics converts the position into joint position, which is tracked by low-level PD. The lower branch (red block) is only used in physical deployment (Section 3.5).

### 3.1. Learning-Based Stance Control

#### 3.1.1. Observation Space and Action Space

Observation space $o_{t}$ consists of 33 dimensions of proprioceptive information. The observation vectors are normalized before being given to policy to neutralize the amplitude difference between various observation channels (For normalization parameter update strategy, see Section 3.1.4). Its components are as follows.
(1)$$o_{t}=\left\{q_{b,t},q_{b}^{*},\omega_{t},v_{t},\omega^{*},v^{*},e_{t},a_{t-1}\right\}$$

Action space $a_{t}$ consists of 6-dimension expected GRF which includes reaction force *F* and torque τ in XYZ directions. $F_{y}$ component of action is mirrored along zero at the left stance phase so that the policy network only has to explore a simpler region rather than two unconnected regions. Its component is as follows.
(2)$$a_{t}=\left\{F_{x},F_{y},F_{z},\tau_{x},\tau_{y},\tau_{z}\right\}$$

#### 3.1.2. Reward Design

Reward function design is also vital for training. Improper reward function would lead to reward hacking, a phenomenon where policy takes unintended behavior to obtain high reward value. Therefore, reward value should be a comprehensive evaluation of task performance and meanwhile dense and smooth enough to learn. For locomotion tasks, the reward function mainly consists of tracking terms that follow a goal and stabilizing terms that prevent falling or collision. In our work, the reward function is composed of three objectives including balancing, command tracking, and gait regulating. The full reward composition is shown below.
(3)$$\begin{array}{ccc} R_{track} & = & F_{\alpha_{1},\beta_{1}}\left(|\right|v_{t}^{*}-v_{t}\left||\right) \end{array}$$
(4)$$\begin{array}{ccc} R_{pose} & = & F_{\alpha_{2},\beta_{2}}\left(1-\xi_{t}^{*T}\cdot \xi_{t}\right) \end{array}$$
(5)$$\begin{array}{ccc} R_{height} & = & F_{\alpha_{3},\beta_{3}}\left(\right|p_{z,t}^{*}-p_{z,t}\left|\right) \end{array}$$
(6)$$\begin{array}{ccc} R_{gait} & = & F_{\alpha_{4},\beta_{4}}\left(\right|c_{t}^{*}-c_{t}\left|\right) \end{array}$$
(7)$$\begin{array}{ccc} R_{slip} & = & F_{\alpha_{5},\beta_{5}}\left(|\right|v_{ankle,t}\left||\right) \end{array}$$
(8)$$\begin{array}{ccc} F_{\alpha ,\beta }\left(x\right) & = & \alpha e^{-\beta x^{2}} \end{array}$$
(9)$$R_{fall}=\left\{\begin{array}{ccc} K & \mathrm{for} & \mathrm{falling}\\ 0 & \mathrm{for} & \mathrm{other} \end{array}\right.$$
(10)$$R_{sum}=R_{track}+R_{pose}+R_{height}+R_{gait}+R_{slip}+R_{fall}$$
where $v_{ankle,t}$ means the ankle velocity of the stance leg. $R_{pose}$ and $R_{slip}$ require the robot to maintain balance with firm steps, $R_{track}$ and $R_{height}$ encourage the robot to traverse under certain commands, and $R_{gait}$ regulates the gait frequency to meet the expected one. All vectors above are expressed in the world frame. $R_{fall}$ penalizes major failures. $F_{\alpha ,\beta }\left(x\right)$ is the Gaussian kernel function, and subscripts α and β are weight coefficients that need to be tuned. Every term except $R_{fall}$ is transformed from negative errors to positive rewards with a Gaussian kernel to prevent proactive termination and set up upper boundaries for each reward term to prevent any term from over-growing and shadowing the others. For specific parameters we used in our experiments, check Table A2.

#### 3.1.3. Policy Representation

In the aforementioned studies [8,9,10,11,12,14,15,19,20,21,26], Actor-Critic framework [34] is adopted to train the policy for locomotion tasks. Multiple layer perception (MLP) with two hidden layers and tanh as activation function is chosen for policy representation in our study because of its simple structure, which boosts training and transition to non-Python machines. Actor and critic networks share the same structure, configuration, and observation input. The policy is trained with the widely-used Proximal Policy Optimization algorithm (PPO) [35] based on surrogate loss, importance sampling, and AdamW optimizer. No experience replay or any buffer of such kind is used in our framework, and all data collected from simulation would be trained only once before being flushed away. The combination of PPO and MLP has already been proved to be feasible in learning robot locomotion policies by multiple research works [11,12,19,20]. Advanced network structures such as LSTM or Transformer can also fit into the proposed framework as policy representation, but more tuning and training effort will be required. The learning rate and its decay threshold have been optimized to accelerate early convergence and avoid over-fitting. Other hyper-parameters are maintained as default values provided by OpenAI Baselines [36]. The detailed hyper-parameters can be found in Table 1 and Table A3.

#### 3.1.4. Adaptive Normalization

Adaptive normalization is applied before the network (the blue block in Figure 2), where the normalization parameters are updated every epoch according to observation storage as shown in equations below and finally converge to the desired values associated with the task so that the system designers do not have to re-tune the normalization module when changing task.
(11)$$\begin{array}{ccc} N_{i} & = & N_{i-1}+n_{i} \end{array}$$
(12)$$\begin{array}{ccc} \mu_{i} & = & \mu_{i-1}+\left({\bar{s}}_{i}-\mu_{i-1}\right)\frac{n_{i}}{N_{i}} \end{array}$$
(13)$$\begin{array}{ccc} \sigma_{i}^{2} & = & \frac{1}{N_{i}}\left[N_{i-1}\sigma_{i-1}^{2}+n_{i}v_{i}+N_{i-1}\sqrt{\frac{n_{i}}{N_{i}}\left(\mu_{i}-\mu_{i-1}\right)}\right] \end{array}$$
(14)$$\begin{array}{ccc} s^{\prime } & = & \frac{s-\mu_{i}}{\sigma_{i}^{2}+10^{-8}},s\in S_{i} \end{array}$$
where $s^{\prime }$ is normalized observation vector. $S_{i}$ is the *i*th batch retrieved from the relevant training episode. $n_{i}$ is the number of samples in batch $S_{i}$. $N_{i}$ is the total number of samples. ${\bar{s}}_{i}$ and $v_{i}$ are the mean value and sample variance of $S_{i}$. $\mu_{i}$ and $\sigma_{i}^{2}$ are the mean value and variance for normalization after the *i*th update.

### 3.2. Curriculum Schema

To improve adaption to various velocity commands and robustness against noisy observations, the policies are trained with a two-stage parallel curriculum as shown in Figure 3.

Before training, parallel environments are built for both accelerating data collection and fostering adaptability. Dynamics randomization is applied to every environment upon setup and remains constant during training. In this way, the policy is exposed to and forced to adapt to a wide range of possible dynamics in training, which reduces the reality gap resulting from dynamics mismatch. Random selection is called in each environment when resetting occurs. It randomly selects a target velocity under world frame from a discrete set. This method is found to be more effective than directly generating continuous random target velocity at every time step. Distribution of random velocity target can be expressed as:(15)$$P\left(v^{*}=i\right)=\frac{i}{2}+0.1,i\in \left\{0.0,0.2,0.4,0.6\right\}$$

In the first stage, the policy is trained in an ideal environment and adaptively updates normalization parameters $\mu_{i}$ and $\sigma_{i}^{2}$. The policy network receives raw observation without any external disturbance. Stage one ends when the average overall reward per trajectory reaches a threshold that guarantees velocity tracking without falling.

In the second stage, we add domain randomization to simulate noisy sensors and external force perturbation on the torso. In test runs of some stage one results, instant failure might happen after epoch time ends. Simulation time per epoch per environment is prolonged by half in this stage to prevent this problem. Normalization parameters become fixed in this stage, in case they are violated by randomization and new failure at the beginning of the new stage.

Detailed parameter configurations are listed in Table A4.

### 3.3. Heuristic Swinging Control

Despite its limitations, heuristics are simple and intuitive enough to be widely used. Inspired by capture point method [37], we adopt a simple linear policy for foot placement planning. The heuristic foot placement policy can be described as follows:(16)$$e_{t}=\delta e_{t}+k_{1}\cdot v_{t}+k_{2}\cdot v_{t}^{*}$$
where *e* is step displacement relative to hip expressed in world frame. δe is an offset vector. $k_{1}$ and $k_{2}$ are both coefficient vectors to be tuned manually.

Foot placement under world frame then has to be converted into joint command. The swing leg is not constrained by contact forces, thus position-based control method can handle it. To avoid collision and singularity, we use trigonometric interpolation in Cartesian space as described below, and transform it into joint space commands via inverse kinematics. We adopted a geometric inverse kinematics computation The planning process is also shown in Figure 4. Joint level position and velocity targets are sent to the low-level controller to be tracked.
(17)$$p_{w}\left(\phi_{t}\right)=\left\{\begin{array}{ccc} A_{1}\cos2\pi \phi_{t}+B_{1} & \mathrm{for} & 0.0\leq \phi_{t}<0.5\\ A_{2}\cos2\pi \phi_{t}+B_{2} & \mathrm{for} & 0.5\leq \phi_{t}<1.0 \end{array}\right.$$
(18)$$\begin{array}{ccc} A_{1} & = & 0.5\times (z_{begin}-z_{mid}) \end{array}$$
(19)$$\begin{array}{ccc} A_{2} & = & 0.5\times (z_{mid}-z_{end}) \end{array}$$
(20)$$\begin{array}{ccc} B_{1} & = & 0.5\times (z_{begin}+z_{mid}) \end{array}$$
(21)$$\begin{array}{ccc} B_{2} & = & 0.5\times (z_{mid}+z_{end}) \end{array}$$
where $z_{begin}$, $z_{end}$, and $z_{mid}$ refer to the z-axis positions of the swing foot when the swing phase begins, ends, and is in the middle. All positions in the equations are expressed in world frame. ϕ refers to the phase value which increases from 0.0 to 1.0 in a gait period.

Phase switching is also done with heuristic conditions. When both the estimated contact force on the swing foot and the time after the last switch reaches their thresholds, or time reaches a larger threshold, a new switch is triggered.

### 3.4. Low-Level Controller

Different low-level controllers are placed after mentioned policies to convert their output signals to joint-level torque commands. For the stance leg, when performing stable locomotion, the acceleration of the robot is trivial enough to be ignored, so that the expected GRF can be approximately equal to the opposite of the end-effect force applied by the stance foot. For the swing leg, a simple proportional-derivative controller calculates feedback joint torque commands. The formulas are defined as follows: (22)$$\begin{array}{ccc} \tau_{stance} & = & J_{stance}^{T}\left(-a_{t}\right) \end{array}$$(23)$$\begin{array}{ccc} \tau_{swing} & = & K_{p}\cdot \left(q_{t}^{*}-q_{t}\right)+K_{d}\cdot \left({\dot{q_{t}}}^{*}-\dot{q_{t}}\right) \end{array}$$
where $J_{stance}^{T}$ is the force jacobian of stance leg, $K_{p}$ and $K_{d}$ are coefficient vectors to be tuned.

### 3.5. Regularized Sim-to-Real Transfer

In addition to randomization mentioned in Section 3.2, dynamics randomization also exists throughout all training stages. Randomized parameters including mass, inertia, and CoM position of links, are generated and fixed when parallel environments are built. See detailed distribution in Table A4.

To maintain the performance of the trained policy, we deploy it with heuristic-based feedback loops to provide regularization on real robots to assist in stabilization around the desired state. Inspired by VMC and Regularized Predictive Control methods, we condition compensation for low-level controllers with floating-base state errors based on heuristic control laws. The feedback formulations are as shown below.
(24)$$\begin{array}{ccc} \Delta F_{z}^{*} & = & K_{z}\left(p_{z}^{*}-p_{z}\right) \end{array}$$
(25)$$\begin{array}{ccc} \Delta \tau_{abduction} & = & K_{y}\left(\theta_{t}^{*}-\theta_{t}\right) \end{array}$$
where $K_{y}$ and $K_{z}$ are both coefficients. $F_{z}$ is expected GRF along Z axis, and θ is the roll angle of the robot base.

## 4. Results

### 4.1. Simulation Training Results

Figure 5 compares the performance of our proposed method with other learning approaches. Shaded areas refer to the range of a group of curves, and the solid curve refers to the average of the group. All approaches share the same hyper-parameters. As for heuristics and low-level parameters, we roughly tuned them to a near-optimal state that would not cause frequent failure and proactive termination behaviors in the first 50 training epochs. We chose the prior-free end-to-end approach proposed by Hurst [14] as the canonical baseline, and an environment that enables domain randomization from the beginning to demonstrate the practicability of the curriculum algorithm.

The proposed hybrid method earns higher rewards on average than the other approaches in both stages. In the early stage of the proposed training method, the curves have large variance because, in each test run, the curriculum stage switches according to its reward level, and the newly introduced disturbance in the second stage causes more failures and reward drop.

In contrast, using end-to-end baseline approach the policy might occasionally perform as nicely as the converged state of the proposed method, but the overall performance is found to be unstable and bares a lower chance of success in practice.

The results of the policy without the curriculum are shown in orange. Corresponding curves reveal relatively weak performance with a high risk of falling. The possible reason is that an over-challenging environment at the beginning leads to frequent failure, then lack of valid data to update policy and normalization parameters, and eventually early convergence before the policy learns to maintain balance.

Figure 6 shows the tracking performances of horizontal velocity. In the 40 s test episode, the velocity reference is composed of a series of step signals. Compared with end-to-end training results, the hybrid policy is proved able to track velocity reference up to 2 km/h with less tracking error. The mean squared error of the hybrid policy and end-to-end baseline policy are $2.232\times 10^{-3}$ and $1.282\times 10^{-2}$. We also tested the same trained hybrid policy using robot models with 5 kg heavier torso (purple curve) and 5 kg lighter torso (red curve). Generally, although payload variations cause larger oscillations and slower responding performance, the proposed hybrid policy shows resistance to mass and inertia discrepancy and finishes tracking tasks without falling.

Figure 7 shows the phase plane plot of X velocity tracking error using different policies. All curves contain data of 2 s since velocity references shift. The curves of the hybrid policy (orange and red) display a spiral pattern attracted to a region centering (0, 0). The stability of the proposed hybrid method is displayed. The end-to-end baseline policy converges to an offset region with a relatively random pattern, which suggests larger tracking error and less stability.

Figure 8 shows the comparison of contact points and joint-level states. Trained hybrid policy produces a regular gait with a frequency of 2.1 Hz, and steady contact indicates firm contact within the stance phase. Compared with the hybrid policy, the baseline policy’s performance is less periodic for its spiky joint angle curves and fitful contacts during the left stance phase.

Figure 9 displays how the robot controlled by hybrid policy regains its balance when being externally pushed. A lateral pushing force of 14 N in world frame is randomly applied on a point on its torso for 0.5 s when the robot is walking 0.6 m/s. After the pose is dragged away from the upright state, two horizontal steps are then taken by the heuristic policy, meantime learned policy maintains the posture of the torso. It is proven that the trained hybrid locomotion controller has enough robustness to endure pushing force and retain balance. In contrast, end-to-end baseline policy cannot endure the same pushing force and falls with brittle actions. In consideration of safety for humans and robots, we did not transfer trained baseline policy onto real robots.

### 4.2. Sim-to-Real Transfer

Figure 10 shows snapshots of a real Wukong-IV robot controlled with the trained hybrid policy. The robot can maintain balance and step in place. From Figure 11 it is clear that equipped with assistive feedback, the performance of the proposed hybrid policy on a real robot is more stable than ones without extra feedback loops. Unaided policies produce vigorous oscillations and lead to failure soon after.

To prove that assistive feedback is not overwriting the policy, in other experiments, the policy network was disabled and assistive feedback was in charge of actuating the robot. As expected, despite multiple fine-tuning being carried out, the robot still instantly fell and crashed.

## 5. Discussion

We have presented a hybrid locomotion policy to produce nice results with achieving resilient bipedal walking. Learning and heuristics cooperate within the proposed structure to finish the locomotion task. In the simulation, the policy is capable of tracking velocity command up to 0.6 m/s. Sim-to-real transfer to Wukong-IV is feasible with assistive feedback loops. The proposed hybrid framework yields higher reward and better performance than similar methods in terms of balance keeping and command tracking accuracy, and thus proved to be a stable and efficient approach to training a locomotion controller with adaption and robustness.

However, there are also limitations in the proposed methodology. Firstly, the performance of policies on a real robot is degraded compared with it in the simulation. The most possible reason is that the simulation and solution results of the software are too ideal, resulting in a large gap with reality. During deployment, policies cannot fully bridge the sim-to-real gap even with assistive feedback loops. In order to improve the performance of real robots, more advanced accurate identification work should be done, and detailed research on more oriented domain randomization combined with the dynamics of legged robots will be demanded in the future. Secondly, the proposed locomotion is still a blind one and just works on flat ground. For a humanoid robot, a highly nonlinear system, relying solely on proprioceptive control, is not conducive for the robot to face complex terrain. Future work is also recommended to adapt to more challenging terrain such as stairs and uneven surfaces with perceptual information.

## Figures and Tables

**Figure 1 micromachines-13-01688-f001:**
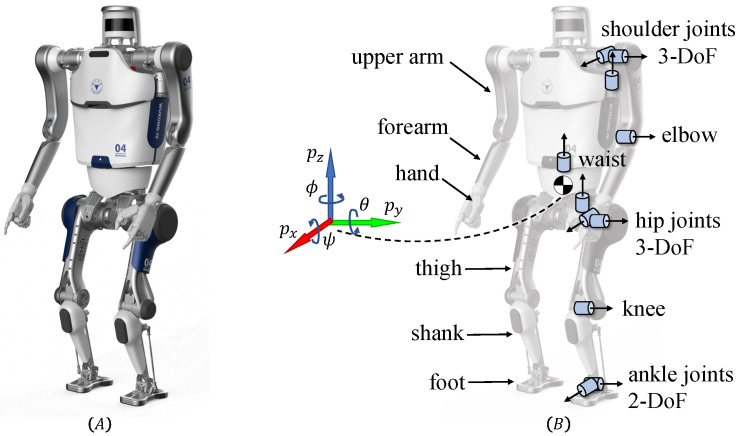
(**A**) Picture and hardware description of Wukong-IV humanoid robot. (**B**) Diagram of joint and link definition for Wukong-IV humanoid robot abstract model.

**Figure 2 micromachines-13-01688-f002:**
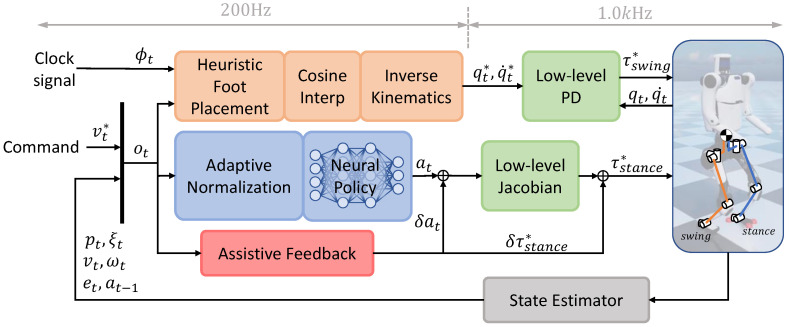
Control framework of hybrid walker. Locomotion task is divided into two different phases and managed by two separate branches. Heuristics swing controller (orange blocks) controls swing leg with a low-cost position-based approach. Learning-based stance controller (blue blocks) adopts NN policy for controlling. Assistive branch (red block) is active only on the real robot for compensating reality gap. Clock signal ϕ is a periodic ramp signal that increases from 0.0 to 1.0 throughout a gait period.

**Figure 3 micromachines-13-01688-f003:**
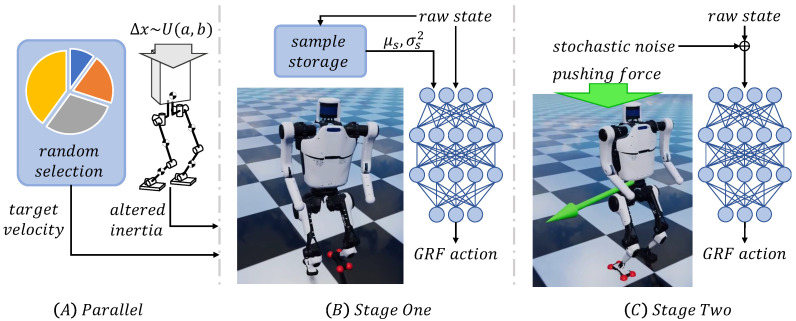
Curriculum schema. Section (**A**) shows parallel randomization curriculum including mass, inertia and task goal randomization. (**B**) shows first curriculum stage with ideal training environment. (**C**) describes second curriculum stage with sensor noise and random force perturbations. Check Section 3.2 for more details.

**Figure 4 micromachines-13-01688-f004:**
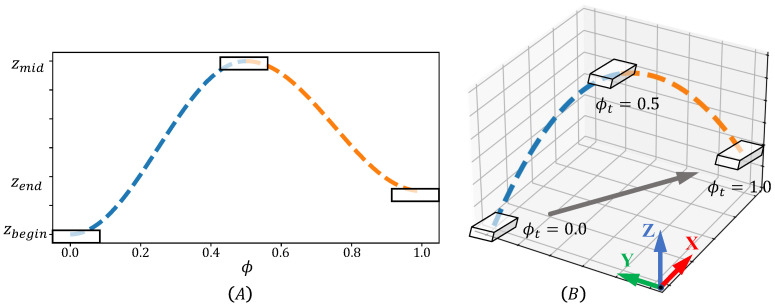
Planned Foot Swinging Trajectory. (**A**) is the plot of phase value and Z axis position of swing foot. White rectangle demonstrates the target position of foot under specific phase value ϕ. (**B**) shows the 3-D curve of the planned spatial trajectory.

**Figure 5 micromachines-13-01688-f005:**
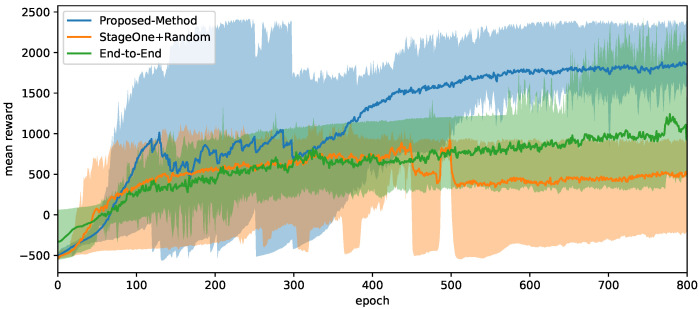
Training reward curves produced by different approaches. Vertical axis represents the average reward gained in one episode among all environments. Blue: proposed approach. Orange: hybrid method without curriculum stage one. Green: end-to-end aproach.

**Figure 6 micromachines-13-01688-f006:**
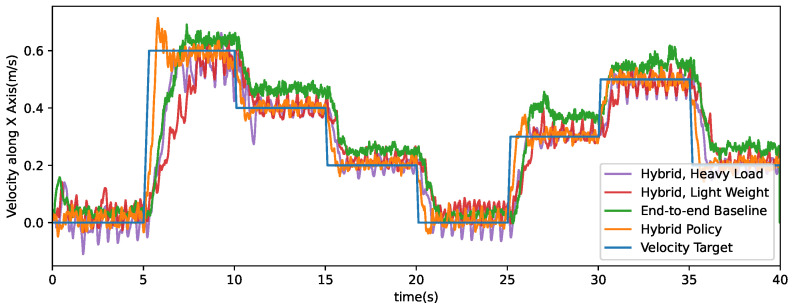
Step respond to step references. Data produced by a simulated robot controlled by different policies to track a predefined reference command signal composed of multiple step signals.

**Figure 7 micromachines-13-01688-f007:**
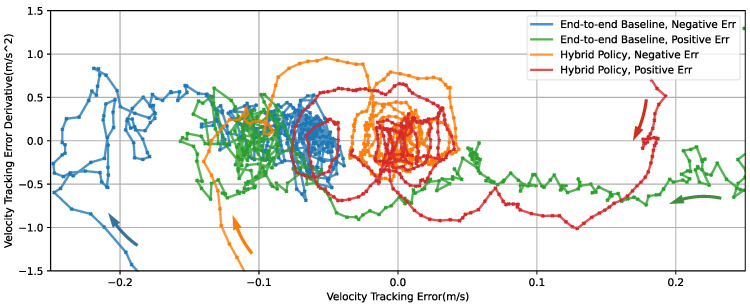
Phase plot of velocity X tracking error. Arrows in different colors indicate the direction of corresponding phase curve. Data obtained from slicing the simulation trajectory in Figure 6.

**Figure 8 micromachines-13-01688-f008:**
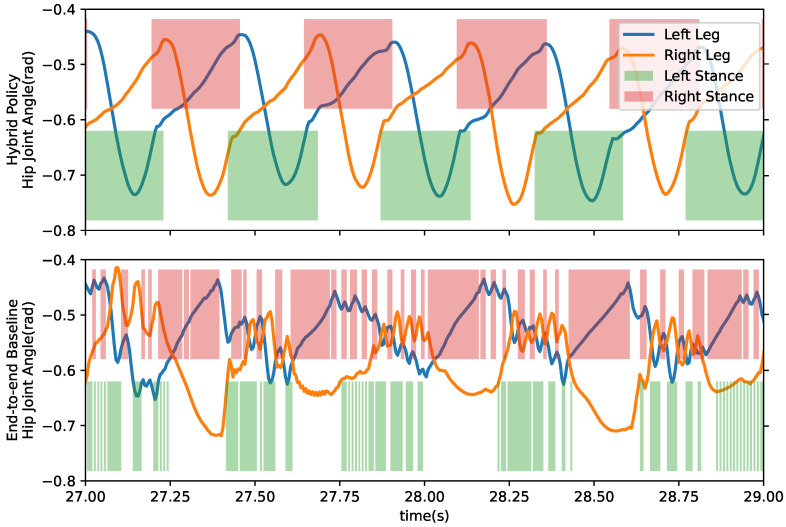
Plot of hip joint position and contact time series while walking. Colored area refers to the foot is having contact with ground at the time. Upper sub-plot’s data is generated by proposed hybrid method, and lower sub-plot plots end-to-end baseline data. Both trajectories are recorded from the test episode described in Figure 6.

**Figure 9 micromachines-13-01688-f009:**
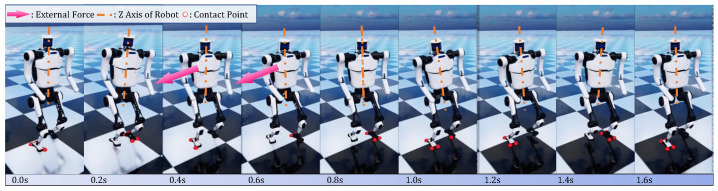
Snapshots of recovering from pushing force. Orange dashed lines refers to the Z axis of robot frame (Corresponding video of the snapshots can be found in Appendix A).

**Figure 10 micromachines-13-01688-f010:**
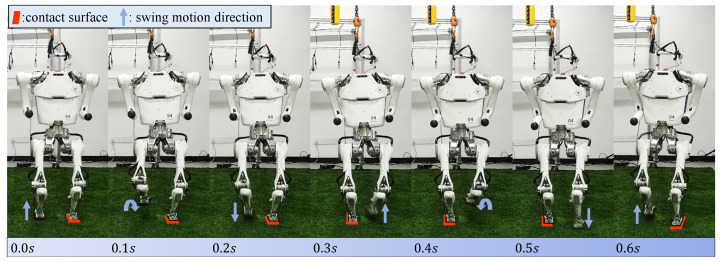
Snapshots of real Wukong-IV robot stepping in place. The trained policy is the same with policy tested in Section 4 (Corresponding video of the snapshots can be found in Appendix A).

**Figure 11 micromachines-13-01688-f011:**
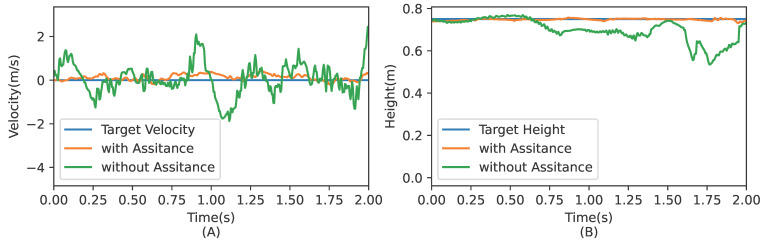
Plots of real robot with or without proposed assistive feedback loop. (**A**) Real robot velocity plot when stepping in place. (**B**) CoM height plot when stepping in place. Orange plots refers to the same experiment as shown in Figure 9. Green plots refers to hybrid policies without assistive feedback.

**Table 1 micromachines-13-01688-t001:** Neural Network Hyper Parameters

Parameters	Value
Network Type	MLP
Latent Layer	2
Latent Node Number	256
Activation	Tanh

## Data Availability

Not applicable.

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
