# Peer review of "Hybrid Bipedal Locomotion Based on Reinforcement Learning and Heuristics"

_micromachines, 2022, doi:10.3390/mi13101688_

Round 1

Reviewer 1 Report (Previous Reviewer 1)

The topic is relevant to the journal. However, this paper is pretty general and lack of details and rigorous discussion. This paper should have more theoretical contributions. I would suggest major revision based on the following comments:

1/ The abstract must be expanded, show the main contribution, in the abstract put a short description of the results obtained (precentage, comparative of errors, etc.)

2/ Each process in Figure 1 is necessary to mention, for example; some contents in Blocks: Inverse Kinematics, Low-level PD, and Adaptive Normalization for better clarification of the proposal. There is only a very general block diagram but detailed designs, developments, and experimental studies were not explained in detail in Figure 1. Adding a clearer explanation to the figures will help the reader understand more clearly what the objective function is in each of the examples used.

3/ In this paper, where the mechanical model is described, it would be interesting to see a diagram with the elements of the robot model.

4/ Please clarify it the control design is indeed model-free and requires no information about system dynamics.

5/ The reviewer has a major concern regarding the motivation for the use of reinforcement learning. The author may discuss how reinforcement learning will be suitable to solve this problem. In-depth studies of neural network identification methods to exploit the data are not given. You could better highlight the architecture and training algorithm. What novelties you have introduced in the structure of reinforcement learning?

6/ The controller design is not clear. How about the stability of the designed controller?

7/ In simulations, the parameters are set to certain values, please add more details of how the parameters of the controller are obtained. How the full-state constraints are chosen? It is better to explain how the values of the control parameters in the proposed method are adjusted? Whether these parameters are optimal for simulation results? More explanation and evidence should be given in detail.

8/ Please add a few phrases to emphasize the usefulness/relevance of the training data (in case I understood it right).

9) I guess that you are optimizing parameters, but it is not clear, please give more details.

10) Simulation results in figures 6-10 are not compared with results obtained for other solutions. More simulation results and a formal comparison of results are needed. The reviewer would like to suggest the authors compare, if possible, their results with some recently published work and clearly show the new design features in the current work.

11/ To examine the validity of the proposed approach, disturbance and load effects should be included to examine the effectiveness and validity of the approach.

Author Response

Please kindly check the attachment for point-wise response.

Reviewer 2 Report (New Reviewer)

# General Comments

The paper presents a control method that combines DRL and heuristic policies. They also rely on domain randomization during training. Experiments are performed in simulation and validated on a real robot.

Overall, the paper is difficult to understand. The introduction, in particular, is not easy to read or understand. Here are some examples of grammatical errors found throughout the entire paper:

- line 13 "are developed" -> "have been developed"
- line 55 "A common issue such summation mechanism is facing..." -> "A common issue facing such summation mechanisms is..."
- add spaces before opening parentheses
- Line 106 "The dimensions of observation space..." -> "Observation space dimensions..."
- line 176 "Despite of the limitations..." -> "Despite its limitations..."

# Specific Comments

(line 9) "...the proposed hybrid approach performed well..."
Please be more specific here. Does the approach match, exceed, or approach current methods?

(figure 1) Please describe symbols (clock and gamepad) in the caption or using a legend. Math notation should be introduced in text closer to the figure. The text description of figure 1 also needs to provide more detail about the various blocks.

(figure 4) This figure should not be referenced prior to figures 2 and 3. Either reorder the text or reorder the figures.

(line 122) Define "reward hacking."

(Line 137) "In the aforementioned studies"
How far before were they mentioned? Please specify the studies again.

(line 138) "Multiple layer perceptron(MLP) with two hidden layers and tanh as activation function is chosen for policy representation in our study because of its simple structure, which boosts training and transition to non-Python machines"
What do you mean by boosts training? I assume training time?
What do you mean by boosts transition? I assume the ability to export the model and import into a non-Python runtime? What framework is used for training?

(line 144) "23?"

(line 145) "Advanced network structures can also fit into proposed framework as policy presentation, but respective tuning will be required."
What is respective tuning?

(figure 2) Please elaborate more about this figure in the caption. Walking through the process would be useful.

(equation 14) Some equations include time as a subscript and some do not. It is not clear when and why the distinction is made.

(220) "... listed at 2." -> "... listed in Table 2."

(figure 5) What is the relationship between reward and walking ability? Is there a point of diminishing returns?

(figure 6) It would be good to add a video corresponding with the given curve. Both for simulation and the real system.

(figure 6) Compare with tracking plots for the two comparison methods. How many seconds before they fall?

(figure 6) How random is the process? Does the proposed method also fall from time to time?

(figure 8) This figure does not appear overly useful.

(figures 9 and 10) The real robot only walks for 2 seconds?

(supplementary video) The real-world video clips are too short and it is difficult to understand the behavior from them.

Author Response

Please kindly check the attachment for point-wise response.

Round 2

Reviewer 1 Report (Previous Reviewer 1)

The revision is satisfied to the reviewer and it is suggested to be accepted.

Author Response

For point-wise response to the comments, please see the attachment.

Reviewer 2 Report (New Reviewer)

# General Comments

The draft still contains many grammatical errors and stylistic inconsistencies. I mentioned a few specific ones in my first round review that I still see. For inconsistencies, I see quite a few brackets with and without a preceding (or ending) space. I recommend significant editing to the text.

# Specific Comments

(figure 1) Add the notations listed in section 2.2 to this diagram.

(line 119) What is meant by "compensation of the quantity"?

(equation 10; table A2) Did the authors try different coefficients? Also, the F(.) function should have a subscript since it is different for each reward component.

(figure 7) The colors make this figure difficult to understand. Ensure that the points are the same color as the matching line, and maybe add line styles.

(line 274) Please provide a frame of reference for 14N.

(figure 10) Please add annotations to the images in figure 10 so that it is easier to understand the point being made---that the robot can step in place.

Author Response

For point-wise response to the comments, please see the attachment.

This manuscript is a resubmission of an earlier submission. The following is a list of the peer review reports and author responses from that submission.

Round 1

Reviewer 1 Report

In my opinion, the topic of this paper is of interest and it is a study to deal with a necessary problem, however, it lacks research elements, there are some things that need to be addressed to meet the quality publication. I would suggest major revision based on the following comments:

1/ The problem is interesting but the description of the methodologies is superficial and it is hard to judge the relevance and new insight provided by the paper. It is rather an implementation report than a scientific research paper. The major concern is that this paper is pretty general and lack of details and rigorous discussion. This paper should have more theoretical contributions. Only a structural framework for engineering is provided, but theoretical framework is very poor.

2/ In the introduction, it is suggested that the novel index of this paper should be explained in detail. And the introduction should be added to do a better job of explaining the existing methods and why they are or are not valuable. The authors reviewed and summarized some existing methods relating to their work. Nonetheless, they have not clearly highlighted what novelty they have proposed in the manuscript. The novelty of the manuscript should be highlighted more specifically in this part.

3/ Write the organization of the paper in the introduction part.

4/ Some abbreviations, such as CPG, PD, PPO and DOFs in the paper, should be spelled out when they are introduced.

5/ Please highlight which section(s) discusses each of the contributions. This way there will be cohesiveness in the manuscript contents.

6/ There is only very general block diagram but detailed designs, developments, and experimental studies in Figure 1. The statistical data given are not persuasive. Some contents in Blocks: Inverse Kinematics, Low-level PD, Adaptive Normalization… were not mentioned in the manuscript.

7/ Some sub-sections in Section 2 are not clear. In-depth studies of machine learning to exploit the data are not given. You could better highlight the architecture and training algorithm. What novelties you have introduced in the structure of the reinforcement learning. The authors just showed the general structure but not detailed designs. The authors have to show the manner of implementation in professional detail which will be beneficial to the readers. A detailed explanation of proposed reinforcement learning needs to be explained. How about the structure or configuration of reinforcement learning? A figure shows the relationship between layers and architecture of the proposed method? I think that it is better to explain how the author designs layers and how to decide the number of neurons in each layer? The author may discuss how proposed reinforcement learning will be suitable to solve this problem.

8/ Which evaluation method that the authors used in the model (cross-validation or train/val split)? Based on the evaluation method, the authors should clearly show that the presented performance results were training or testing set.

9/ Hyperparameter optimization is a very important step in any deep learning problem. How did the authors deal with this problem in this study?

10/ Detailed implementation information should be provided (hardware, software, configuration, settings). A detailed discussion of hardware and software applied to the system should be mentioned. Provide specifications of the hardware and software used for simulation of the approach. Because there is not enough data on this manuscript, the research results on the core idea of this manuscript seem unreliable.

11/The reviewer would like to suggest the authors compare, if possible, their results with some recently published work and clearly show the new design features in the current work. More discussions should be given to clearly demonstrate the limitations/validity of the obtained results.

12/ Where is the conclusion part of this manuscript? The Conclusion section is superficial, should include quantitative results, advantages and disadvantages, limitations and recommendations for new implementations and future work.

13/ The manuscript writing can be further polished with professional English, some typos and grammatical errors should be checked carefully, and some formatting problems that need to be modified.

Author Response

No.: micromachines-1833825

Manuscript: Hybrid Bipedal Locomotion Based on Reinforcement Learning and Heuristics

***************************************************************

Responses to Reviewers’ Comments:

Reviewer#1’s comments:

In my opinion, the topic of this paper is of interest and it is a study to deal with a necessary problem, however, it lacks research elements, there are some things that need to be addressed to meet the quality publication. I would suggest major revision based on the following comments:

1/ The problem is interesting but the description of the methodologies is superficial and it is hard to judge the relevance and new insight provided by the paper. It is rather an implementation report than a scientific research paper. The major concern is that this paper is pretty general and lack of details and rigorous discussion. This paper should have more theoretical contributions. Only a structural framework for engineering is provided, but theoretical framework is very poor.

Response:

First of all, thank you for your suggestions. Previously, some of comparison is not fully described in our introduction and methodology part, and some details of implementation are left due to page restriction. We have extended the introduction, discussion and theoretical part and included more comparation and analysis. Instead of the commonly adopted end-to-end method that control the robot throughout the period with a unified neural network, we divided the task into different phases and combined control algorithms according to the phase’s attribute, and proposed the structure.

2/ In the introduction, it is suggested that the novel index of this paper should be explained in detail. And the introduction should be added to do a better job of explaining the existing methods and why they are or are not valuable. The authors reviewed and summarized some existing methods relating to their work. Nonetheless, they have not clearly highlighted what novelty they have proposed in the manuscript. The novelty of the manuscript should be highlighted more specifically in this part.

Response: Thanks for reminding. We have clarified the details of related works and the difference between those methods and our proposed method. The edited parts are shown in red fonts. Existing learning-based methods that combine model-based control usually adopt a summation manner that might cause constraining or overwriting conflicts. The proposed method of ours combined heuristics and learning-based methods with assigning them to different phases, thus reached cooperation while avoiding conflict. Out results also proved that proposed methodology can yield better locomotion policy than existing ones.

3/ Write the organization of the paper in the introduction part.

4/ Some abbreviations, such as CPG, PD, PPO and DOFs in the paper, should be spelled out when they are introduced.

5/ Please highlight which section(s) discusses each of the contributions. This way there will be cohesiveness in the manuscript contents.

Response:

All the advised parts have been added. It really makes the article more structured. As for these abbreviations, some words were used so often in our work so we did forget to give a full interpretation. According to the format of the journal provided, abbreviation section in the end of the text and the missed abbreviations are added there. To prevent the mail text from being redundant, we settle full spelling there. Please check appendix for details.

6/ There is only very general block diagram but detailed designs, developments, and experimental studies in Figure 1. The statistical data given are not persuasive. Some contents in Blocks: Inverse Kinematics, Low-level PD, Adaptive Normalization… were not mentioned in the manuscript.

Response:

Personally we believe that Inverse Kinematics is a common technique in robotics that convert the coordinate from task space to joint space. It's tightly bonded to joint design of a specific robot so we think too detailed mathematical calculations should not be included in a block diagram. The word ‘Inverse Kinematics’ is enough.

Proportional Derivative Controller is also a commonly used low level model-free controller that can stabilize the joint.

As for Adaptive Normalization it means re-calculate the mean and standard deviation of collected data and update the parameters for normalization. Related information is mentioned in Curriculum Training Section.

7/ Some sub-sections in Section 2 are not clear. In-depth studies of machine learning to exploit the data are not given. You could better highlight the architecture and training algorithm. What novelties you have introduced in the structure of the reinforcement learning. The authors just showed the general structure but not detailed designs. The authors have to show the manner of implementation in professional detail which will be beneficial to the readers. A detailed explanation of proposed reinforcement learning needs to be explained. How about the structure or configuration of reinforcement learning? A figure shows the relationship between layers and architecture of the proposed method? I think that it is better to explain how the author designs layers and how to decide the number of neurons in each layer? The author may discuss how proposed reinforcement learning will be suitable to solve this problem.

Response:

Thanks for the suggestions. It makes sense to explain further. We adopted a minimal structure of policy representation, which is a MLP trained with PPO. In corresponding section(S2.1.3) we explained why we chose MLP because it’s easy for our deployment since only C++ works on real robot. We also hope that implementing the learning architecture won't be an obstacle for other teams to reappear our results. Hence the learning architecture is not specially designed and barely a minimal option. We have added some explanation about our reinforcement learning framework.

PPO algorithm has been widely used in previous study (e.g. reference 9, 10, 17, 18) and their results proved its performance and feasibility, so it’s reasonable for us to choose PPO.

8/ Which evaluation method that the authors used in the model (cross-validation or train/val split)? Based on the evaluation method, the authors should clearly show that the presented performance results were training or testing set.

Response:

As mentioned above, since there is no experience replay mechanism in our learning structure, the policy is always facing an interactive environment instead of a pool of collected trajectories. In other word, our training data is generated in the training process by interaction between the agent and the simulation enviroment. In this case the action generated by the policy would instantly influence the state of next time step and alter the choice range of states.

Every time the policy is facing a newly build world so we consider all the result to be testing.

9/ Hyperparameter optimization is a very important step in any deep learning problem. How did the authors deal with this problem in this study?

Response:

In our vision, hyperparameters should not be a problem worth spending too much time on in this work. We are dedicated to reduce effort spent on tuning as much as we can. We believe that a proper framework can make tuning work trivial. In this study we just made sure that the hyperparameters are all in the correct scales. For example, the coefficients of the reward terms are made sure not to outweigh the others while having similar gradients. This condition is sufficient for the PPO algorithm to converge the network to a desired one.

10/ Detailed implementation information should be provided (hardware, software, configuration, settings). A detailed discussion of hardware and software applied to the system should be mentioned. Provide specifications of the hardware and software used for simulation of the approach. Because there is not enough data on this manuscript, the research results on the core idea of this manuscript seem unreliable.

Response:

All software and hardware are already mentioned in 'platform overview' section (S3.1) with references to find them. In software, RaiSim and PyTorch are powerful enough to cover all the demand to carry out the research. In hardware, Wukong robot is built by our research team and do not rely on extra software for interfacing. Table A5 in Appendix B shows its hardware specifications. All code on the real robot are implemented with C++ under Ubuntu 16.04.

11/The reviewer would like to suggest the authors compare, if possible, their results with some recently published work and clearly show the new design features in the current work. More discussions should be given to clearly demonstrate the limitations/validity of the obtained results.

Response:

The comparison was made at Figure 5. The "End-to-End" plot is generated with method in " Learning agile and dynamic motor skills for legged robots (2019)" and " Sim-to-real: learning agile locomotion for quadruped Robots (2018)". But the policies trained with this method all failed to perform stable locomotion and thus were not further tested in sections below. The unstable locomotion task of bipedal locomotion could hardly be trained using the straightforward rewards. A short explanation can be found in the ‘Simulation Training Results’ section (S3.2).

12/ Where is the conclusion part of this manuscript? The Conclusion section is superficial, should include quantitative results, advantages and disadvantages, limitations and recommendations for new implementations and future work.

Response:

We put these summary contents in the discussion part. Because the guidelines provided by the journal mentions "Conclusion section is not mandatory, but can be added to the manuscript if the discussion is unusually long or complex." But you are right, the discussion is not informative enough. We have enriched the discussion with further details and analysis.

13/ The manuscript writing can be further polished with professional English, some typos and grammatical errors should be checked carefully, and some formatting problems that need to be modified.

Thanks for the suggestions. Some language issues have been fixed in the revised version.

Reviewer 2 Report

I like the work on bipedal robotic locomotion with having a sufficient amount of model-based perspective and certain RL techniques. However, with the state of the art, it is very hard to understand what is the contribution of this paper. Many discussions of the approach were barely scratching the surface; comparisons shall be discussed better with the canonical approaches and many RL work. I also don't see a video of the results. Video demonstrations are very important to have in this field.  I have several general suggestions: 

- Talk more about the robot design. Right now, the paper only focuses on the control approach. To be honest, it's very hard to see how this is better/different from many RL+model based methods. With a focus on designing controllers on a custom-designed robot, it can sell better. 

-Focus on presenting the benefits of this proposed approach compared to the existing ones. 3D bipedal walking is no longer new in 2022. We need to know better about the pros and cons across different methods

- Upload a video of the experiments. This is very important. 

Technical questions; 

- How would the accuracy of the state estimator affect the performance of this work?

- (2) on the footplacement resembles many stepping controllers from the SLIP model (Raibert), step-to-step dynamics based. Do the values of k1 k2 turn out to be similar to the ones from LIP based stepping controllers, e.g. step-to-step based stepping controllers?

- Does the trajectory design of the swing foot affect the walking performance?

- The low-level controller appears to be very out-dated. Would the authors be able to apply any TSC/CLF QP based controllers. This is not required. Also, can we include the PD gains the RL for learning better PD gains rather than hand-tuning. 

- the sim-to-real transfer has a heuristic term. It's not clear to me what are the purposes of having them. Why cannot formulate that in the reward of the RL. 

- Fig. 5, the proposed method has a very large variation in the beginning of training. Why is that?

Some minor problems:

-Heuristics is REALLY not a formal/principled terminology. Please better wording. 

- Paragraph 4, Jonathan [11-13], the way of reference is not formal. First name shall not be used. 

- The citations are very RL heavy. Many papers are just from the two groups on OSU and ETH. The authors should include a more diverse and comprehensive set of literature. Consider including some state-of-the-art model-based methods and robots. 

Author Response

No.: micromachines-1833825

Manuscript: Hybrid Bipedal Locomotion Based on Reinforcement Learning and Heuristics

***************************************************************

Responses to Reviewers’ Comments:

Reviewer#2’s comments:

- Talk more about the robot design. Right now, the paper only focuses on the control approach. To be honest, it's very hard to see how this is better/different from many RL+model based methods. With a focus on designing controllers on a custom-designed robot, it can sell better. 

Response: Thank you for the suggestion. We have introduced more about our humanoid robot “Wukong-IV” in the platform overview part. Meanwhile, detailed information about advantages of proposed methodology has been added to the introduction part. The proposed hybrid method provides a phase-specific method that activates controllers with corresponding features in different phases. It prevents the risk of conflict and overwritten in summation of learning and model-based approaches. In addition, the whole structure is efficient and light-weighted without effort to iterate for numerical solutions. In terms of structure-related controller design, our robot is a bio-inspired humanoid, and we have added related explanations in the introduction part.

-Focus on presenting the benefits of this proposed approach compared to the existing ones. 3D bipedal walking is no longer new in 2022. We need to know better about the pros and cons across different methods

Response: Thank you for the advice. Comparison is indeed vital to display advantages. We have enriched comparisons between end-to-end approaches and summation approaches. For methods that we do not have enough conditions to do quantitate comparisons yet, we made qualitative comparisons. Added parts are highlighted in the main text.

- Upload a video of the experiments. This is very important. 

Response: Thanks for the advice. We have organized our test results as a short video. It will be upload along with this revision.

Technical questions; 

- How would the accuracy of the state estimator affect the performance of this work?

Response: On real robots, ground truth states cannot be obtained, and the only way to measure is through state estimator. Due to sensor noise and residuals in numerical solving algorithm, state estimators would inevitably have offsets and variations. According to our observation, the most important states for the trained policy is base velocity and base orientation. Most our trained policies are sensitive to these states, and might perform unexpected actions when received noisy input signal. As a countermeasure, we adopted domain randomization and add sensor noise after the policy learned to perform basic task. We ensured the magnitude of sensor noise can fully cover the possible difference between simulator and real-world estimators.

- (2) on the footplacement resembles many stepping controllers from the SLIP model (Raibert), step-to-step dynamics based. Do the values of k1 k2 turn out to be similar to the ones from LIP based stepping controllers, e.g. step-to-step based stepping controllers?

Response: In our experiments, swing controller parameters were tuned based on Raibert-style controllers. We first tuned in simulation to find a set of parameters for the swing leg that works best with the trained policy. Then we used the same parameters to control the real robot. We observed that the value of step target, which is K1 and K2, have the same optimal point with Raibert-style controller on our robot. We adopted those parameters for generating the test results shown in the paper.

- Does the trajectory design of the swing foot affect the walking performance?

Response: Swing trajectory is an essential module in the locomotion control system. We believe improper swing trajectory design is detrimental to locomotion performance, such as heavy impact and violent foot lifting. These behaviors would influence the overall momentum of the robot system and cause model mismatch and policy failure even if the robot’s legs are relative light-weighted. In our methodology we adopted trigonometric curve for smooth trajectory. We believe that using Bezier-based curve generator may have similar results.

- The low-level controller appears to be very out-dated. Would the authors be able to apply any TSC/CLF QP based controllers. This is not required. Also, can we include the PD gains the RL for learning better PD gains rather than hand-tuning. 

Response: It’s a great idea to upgrade the low-level controllers with better performances. We adopted low-cost PD controller and Jacobian in or methodology, and they are sufficient for performing stable locomotion.  We have no doubt that introducing advanced low-level controllers will improve the robustness of the system, and it’s also a future orientation of our team. We are planning to investigate that if better low-level components can reduce the sim-to-real effort. As for tuning PD gains with RL, it is a plausible method to increase the generalization ability. We didn’t integrate the technique in case of being redundant since there have been other works about this topic, e.g. Ames et al.[1] tuned the gains with preference-based learning.

- the sim-to-real transfer has a heuristic term. It's not clear to me what are the purposes of having them. Why cannot formulate that in the reward of the RL. 

Response: Thanks for asking. Extra sim-to-real feedback loops in our control structure are designed for further bridging the reality gap. Although policies are trained to be robust by domain randomizations, imperfect simulators would still cause the gap between simulated and real robots. The feedback loops assist the trained policy in real world to maintain an upright orientation and stable height. The states in extra feedback loop also exists in reward function, and the policy has enough robustness in simulation to maintain balance on its own. The purpose of the extra loops is only to enhance the robustness in the real world.

- Fig. 5, the proposed method has a very large variation in the beginning of training. Why is that?

Response: Thanks for asking. On the reward plot, the proposed method showed massive variance between 100 epochs and 300 epochs. The reason of that is the curriculum learning schema we adopted. In case of overfitting, the stages switches once the average reward value per trajectory reaches a certain threshold. In stage two the policy has to face domain randomization and the reward value would grow again after a sudden drop.

Meanwhile, the policy for stance control is such a very small-scale feedforward network that the initial weights generated by different random seeds would dramatically influence convergence performance. Therefore, test runs reach the threshold in various time. Some of the training tests would enter the second stage in their early 100 epochs, while the unluckiest one has to wait for convergence after 300 epochs. In the period between some test runs have reached the threshold and learns to perform under domain randomization with relative low performance, and the others are still converging in stage one and gaining high reward value. That is the source of proposed method’s large variance at the beginning.

Some minor problems:

-Heuristics is REALLY not a formal/principled terminology. Please better wording. 

Response: Thank you for the advice. We have tried to find other word to conclude the bio-inspired methodology such as Raibert’s controller, VAC and VMC etc. when conceptualizing our idea. We ended up following the naming convention in Regularized Predictive Control by Bledt et al[2].

- Paragraph 4, Jonathan [11-13], the way of reference is not formal. First name shall not be used. 

- The citations are very RL heavy. Many papers are just from the two groups on OSU and ETH. The authors should include a more diverse and comprehensive set of literature. Consider including some state-of-the-art model-based methods and robots. 

Response: Appreciations for your suggestion about citations and references. We have improved them accordingly. The citations are now in better formatting.

[1] Csomay-Shanklin, Noel, Maegan Tucker, Min Dai, Jenna Reher, and Aaron D. Ames. "Learning controller gains on bipedal walking robots via user preferences." 2022 International Conference on Robotics and Automation (ICRA), 2022, pp. 10405-10411.

[2] G. Bledt, P. M. Wensing and S. Kim, "Policy-regularized model predictive control to stabilize diverse quadrupedal gaits for the MIT cheetah," 2017 IEEE/RSJ International Conference on Intelligent Robots and Systems (IROS), 2017, pp. 4102-4109.

Round 2

Reviewer 1 Report

Regarding my last round of comments, the authors just tried to clarify what they do but without really addressing the concerns. I still feel that the paper lacks research elements. Only a structural framework for engineering is provided, but the theoretical framework is very poor Authors applied some changes to the paper. However, there are still some comments missing.